# Roles for the RNA-Binding Protein Caper in Reproductive Output in *Drosophila melanogaster*

**DOI:** 10.3390/jdb11010002

**Published:** 2022-12-23

**Authors:** Erika J. Tixtha, Meg K. Super, M. Brandon Titus, Jeremy M. Bono, Eugenia C. Olesnicky

**Affiliations:** Department of Biology, University of Colorado Colorado Springs, 1420 Austin Bluffs Parkway, Colorado Springs, CO 80918, USA

**Keywords:** Caper, RNA-binding proteins, fertility, reproductive output

## Abstract

RNA binding proteins (RBPs) play a fundamental role in the post-transcriptional regulation of gene expression within the germline and nervous system. This is underscored by the prevalence of mutations within RBP-encoding genes being implicated in infertility and neurological disease. We previously described roles for the highly conserved RBP Caper in neurite morphogenesis in the *Drosophila* larval peripheral system and in locomotor behavior. However, *caper* function has not been investigated outside the nervous system, although it is widely expressed in many different tissue types during embryogenesis. Here, we describe novel roles for Caper in fertility and mating behavior. We find that Caper is expressed in ovarian follicles throughout oogenesis but is dispensable for proper patterning of the egg chamber. Additionally, reduced *caper* function, through either a genetic lesion or RNA interference-mediated knockdown of *caper* in the female germline, results in females laying significantly fewer eggs than their control counterparts. Moreover, this phenotype is exacerbated with age. *caper* dysfunction also results in partial embryonic and larval lethality. Given that *caper* is highly conserved across metazoa, these findings may also be relevant to vertebrates.

## 1. Introduction

Post-transcriptional gene regulation is central to the development of both the germline and nervous system [1,2]. RNA binding proteins (RBPs) facilitate various aspects of post-transcriptional gene regulation including alternative splicing, polyadenylation of mRNAs, nuclear export, RNA localization and translational control. RBPs were extensively studied for their roles in *Drosophila* oogenesis and embryonic axis determination. In oogenesis, proper localization and the timing of translation of numerous mRNAs, such as *nanos (nos)*, *bicoid (bcd)*, *oskar (osk)* and *gurken (grk)*, are critical to ensuring appropriate oocyte development [3,4,5,6,7,8]. Indeed, these maternal mRNAs were found to be regulated by myriad RBPs, including Syncrip, Bruno, Staufen, Squid and many others [3,9,10,11]. Similarly, many mRNAs are post-transcriptionally regulated during embryogenesis. The importance of the regulation of localization and translation of these mRNAs is underscored by the fact that their aberrant expression results in abnormalities in polarity and embryonic patterning [12,13,14,15,16]. For instance, when *nos* is inappropriately localized to the anterior of the embryo, an abdomen develops in place of anterior structures [12].

RBPs also have well established roles in neurogenesis. For example, using an RNA interference screen for roles of RBPs in dendrite morphogenesis, we previously found that at least 88 RBPs regulate dendrite formation in *Drosophila* sensory neurons. Furthermore, a subset of these have conserved roles for sensory dendrite development in *C. elegans* [17,18,19,20]. Additionally, many mRNAs, such as *nos*, are also localized to Class IV da neuron dendrites, and their inappropriate localization or translation results in dendritic patterning defects [21,22,23]. Similar to Nos, myriad RBPs are known to play roles in both the germline and the nervous system. For example, the gene *Fmr1* encodes the Fragile X Messenger Ribonucleoprotein (FMRP) whose dysfunction is causative for Fragile X Syndrome, which is characterized by many different aberrant neuronal phenotypes including intellectual disability [2]. However, FMRP dysfunction is also implicated in primary ovarian insufficiency and germline tumor formation in humans [24]. Similarly, the RBP Pumilio (Pum), which was originally identified for its role in *Drosophila* oogenesis, was also implicated in development and function of the nervous system [25,26,27,28]. Indeed, mutations in human *pum* orthologs are associated with primary ovarian insufficiency, as well as adult-onset ataxia and seizures [24,29].

We previously identified the highly conserved RBP-encoding gene *caper* for a role in dendrite and axon morphogenesis of sensory and motor neurons, respectively, in *Drosophila*. Furthermore, while Caper is broadly expressed throughout embryogenesis, including within the germline, nothing is known about Caper function during oogenesis [30,31]. Importantly, *caper* is highly conserved across metazoa, from yeast to *C. elegans* and humans [32]. *Drosophila caper* has two human orthologs, *RBM39* and *RBM23*. RBM39 is expressed in the human ovary and within the ovaries and testes of mice [33,34]. Furthermore, it is ubiquitously expressed in zebrafish during embryogenesis [35]. Here we show that Caper is expressed during *Drosophila* oogenesis and affects viability at multiple developmental stages. Additionally, we find that *caper* plays a role in fecundity, as *caper*-deficient animals develop smaller ovaries and have reduced reproductive output, as compared to controls.

## 2. Materials and Methods

### 2.1. Fly Lines

The *caper^CC01391^* hypomorphic allele utilized in our experiments, referred to here as *caper*^−/−^, was previously described by Olesnicky et al. [30]. The following stocks were obtained from the Bloomington Stock Center: *y^1^ sc* v^1^ sev^21^*; *P{TRiP.HMC03924}attP40*; *UAScaperRNAi^HMC03924^* [36]; *{w[+mC]=Act5C-GAL4}y[1] w[*]*; *P17bFO1/TM6B, Tb[1]*; *UAScaperRNAi ^GLC01382^ y^1^ sc* v^1^ sev^21^*; *P{TRiP.GLC01382}attP2/TM3, Sb^1^*; *P{w[+mC]=UAS-Dcr-2.D}1, w[1118]*; *P{w[+mC]=GAL4-nos.NGT}40*.

Flies for experiments utilizing RNAi lines were maintained at 25 °C with a 12-h light/dark cycle. Experiments utilizing genetic mutations were performed at room temperature unless otherwise stated. Since the *caper^CC01391^* hypomorphic mutant allele was created in a *yw* background, *yw* served as the control [30,37]. For all RNA interference experiments, *Gal4* drivers were outcrossed to *yw*, and the progeny heterozygous for the *Gal4* driver served as controls.

### 2.2. Immunoblotting

Ovaries were dissected on ice from 5 day old *yw* and *caper*^−/−^ females. Lysates were prepared using 20 ovaries and lysing them in 100 ul of lysis buffer (Urea buffer: 0.125 M Tris-HCl pH 6.8, 4% SDS, 20% glycerol, 5 M urea, 0.1 M DTT, 0.01% bromophenol blue). The lysates were produced using a FisherBrand^®^ motorized tissue grinder and RNAse-free disposable pellet pestles. The grinder was used to homogenize the tissue for 30 s, which was then boiled at 100 °C for 5 min. Lysates were centrifuged at 14,000 rpm for 20 min at 4 °C to remove cellular debris. Lysates were then boiled at 100 °C for an additional 5 min and fractionated in Bio-Rad 4–15% Mini-PROTEAN^®^ TGX^TM^ precast gels and transferred to ThermoScientific PVDF membranes by electroblotting at 30 mA overnight at 4 °C. The membranes were then cut between the 50 kDa and 70 kDa marker to detect α-Tubulin (~50 kDa) and Caper (~75–80 kDa). The following primary antibodies were used to incubate for 2 h at room temperature: anti-α-Tubulin 1:25,000 (Sigma-Aldrich T9026) or rabbit anti-Caper1 1:15,000 (GenScript). The following secondary antibodies were used to incubate for an hour at room temperature: peroxidase conjugated to goat anti-mouse IgG, Fc fragment specific (Jackson ImmunoResearch 115-035-008) at a 1:30,000 dilution or peroxidase conjugated to goat anti-rabbit (Abcam ab6721) at a 1:20,000. All antibodies were diluted in 1X TBS, 0.1% Tween 20 and 5% bovine serum albumin (VWR International). Membranes were visualized by chemiluminescence using the Azure Biosystems Radiance Q chemiluminescent substrate with Azure Biosystems C400 imager. Quantification of blots was performed using FIJI (Fiji Is Just ImageJ) Gel analysis tools. The ratio of the detection for Caper compared to the detection of α-Tubulin was used as a form of normalization to account for differences in loading.

### 2.3. Immunofluorescence

Ovaries were dissected on ice from 3–5 day old females in PBS and, subsequently, fixed in 4% paraformaldehyde in PBS for 20 min at room temperature. Ovaries were incubated in Image-iT FX Signal Enhancer (Invitrogen, Waltham, MA, USA) for 30 min at room temperature in the dark and subsequently blocked for 1 h in 5% normal goat serum (NGS) in PBS. Caper 3 antibody [31] was incubated overnight at 4 degrees Celsius at a concentration of 1:250 in 5% NGS in PBS. Alexa flour 546 (Invitrogen) was incubated overnight at 4 degrees Celsius at 1:500.

### 2.4. Fecundity Assay

*caper*^−/−^ and *yw* virgin females were collected on the day of eclosion and set up in four crosses: *caper*^−/−^ females with *caper*^−/−^ males, *caper*^−/−^ females with *yw* males, *yw* females with *caper*^−/−^ males, and *yw* females with *yw* males. In total, 25 vials were set up for each cross, with 5 flies of each sex placed in each vial. Flies were transferred daily into new vials, at which point the vials containing the previous day’s eggs were briefly frozen at −20 °C to arrest development. Eggs were then counted and average output per female was calculated for each vial. The remaining number of living males and females in a vial was recorded, and a vial was removed from the daily averages if all of either sex within it had died. The experiment was terminated when one of the four cross combinations no longer had any vials remaining. Results were verified using the RNAi line *UAScaperRNAi ^GLC01382^* driven by *P{w[+mC]=UAS-Dcr-2.D}1, w[1118]*; *P{w[+mC]=GAL4-nos.NGT}40*. The driver was also outcrossed to *yw* to serve as a control.

### 2.5. Determining Fertilization and Embryonic Lethality

The same four crosses with *caper*^−/−^ and *yw* described above were used for these experiments. For embryo analysis experiments, 50 parent flies of each sex were placed in embryo collection cages. Flies were permitted to lay for about 12 h on apple juice collection plates with a 50:50 mixture of yeast and water, at which point the plates were removed and set aside to age for another 12 h. A 50% bleach solution was then added dropwise to embryos for two minutes to dechorionate them, after which they were fixed for 20 min in a 1:4 solution of 4% paraformaldehyde:heptane. Embryos were then devitellinized with methanol, stained with the nuclear marker DAPI, and scored with a fluorescence microscope to determine the stage in which they arrested development. Given that embryos were allowed to develop from 12–24 h, embryos that failed to develop to at least stage 14 were presumed dead.

### 2.6. Larval and Pupal Lethality Assay

Larvae were collected at the third instar stage, separated by sex, and plated on apple juice plates with a 50:50 mixture of yeast and water available for additional nutrition. Larvae were scored for lethality daily and plates were sprayed with deionized water as needed to avoid desiccation. The remaining larvae were subsequently allowed to pupate and were assessed for lethality at the pupal stage in the same manner.

### 2.7. Mating Assay

Mating experiments were performed as described in [38], with some modifications due to the difference in *Drosophila* species. Virgin females and males for each genotype tested were collected and allowed to age to four days to ensure full reproductive maturity [39]. 15 each of *caper*^−/−^ and *yw* males and females were then placed into the mating chamber, which was an 8″ × 1″ dish that had 4 holes through which an aspirator could extract flies. This experimental setup was utilized 10 times to achieve an adequate sample size. When flies were seen mating, they were observed for 30 s to ensure true copulation, at which point they were extracted via aspirator. When only half of the flies remained in a given chamber, or two hours had passed, the experiment was stopped.

For the *caper* knockdown experiment, the experimental setup was altered so that 30 females of the *UAScaperRNAi^GLC01382^* driven by *nanosGal4*, as described above, were placed in a chamber with 15 *caper*^−/−^ and 15 *yw* males. Then, 30 females of the driver outcrossed to *yw* were placed in a separate, similar chamber with the same number of males as a control due to these females being indistinguishable from the RNAi knockdown females by eye color. Four such chambers were set up for each of these two female genotypes, and arenas were observed for a full two hours. As before, mating pairs were observed for copulation and then removed. Because differences in mating number for each female genotype, rather than mate choice, was the primary focus of this experiment, these assays were not ceased early if half of the flies had mated.

### 2.8. Ovary Size Analysis

Ovaries were dissected on ice for *yw*, *caper*^−/−^, *UAScaperRNAi^GLC01382^* driven by *nanosGal4*, and *nanosGal4 ctl* females at day 3, day 5× and day 14 post-eclosion. Dissected ovaries were immediately placed on a slide and imaged using Brightfield setting at 5 magnification using Leica DM4-B microscope. Ovary measurements were taken in FIJI by drawing a perimeter around the ovary and using the measure tool to calculate the area.

### 2.9. Statistical Analyses

Fecundity data were analyzed using a generalized linear mixed model (GLMM) with a negative binomial error distribution (NB1 parameterization) using the R package ‘glmmTMB’ [40]. The full factorial model included female line, male line and day as factors. Replicate was treated as a random effect to account for repeated measurements from the same set of flies over time. Model fit was assessed using diagnostic plots generated by the R package ‘DHARMa’ [41]. An anova table was generated from model results using the R package ‘car’ [42]. Mortality at different developmental stages was analyzed using GLMs with a binomial error distribution and logit link function. However, the experiment comparing larval survival of *caper*^−/−^ and controls was analyzed using Fisher’s exacts tests because the fact that no female control larvae died hindered estimation of the glm model due to quasi-complete separation. The glm model for embryo mortality included cross as a factor, while larval and pupal mortality analyses included both cross and sex as factors as well as the interaction between these variables. Anova tables were generated from model results using the R package ‘car’ [42]. Post hoc comparisons were analyzed with the R package ‘emmeans’ using Tukey’s adjustment [43]. Male and female mating frequency was analyzed by GLMM with a binomial error distribution and logit link function using the R package ‘lme4’ [44]. Replicate was treated as a random variable.

Ovary size data was analyzed with a full factorial linear mixed effects model using the R package ‘lme4’ [44]. The model included factors for genotype and day and their interaction. Fly id was treated as a random effect since both ovaries were measured for each fly. Anova tables were generated from model results using the R package ‘car’ [42]. Post hoc comparisons were analyzed with the R package ‘emmeans’ using Tukey’s adjustment [43].

## 3. Results

### 3.1. Caper Is Expressed during All Stages of Oogenesis

We previously showed that Caper is widely expressed during embryogenesis, including within the nervous system and within pole cells [30]. To determine whether Caper is also expressed during oogenesis, we performed anti-Caper immunofluorescence on ovarioles derived from *yw* females. We find that Caper protein is expressed throughout all stages of oogenesis including within the germarium. Caper is detected within the nuclei of the cells of the germarium, as is expected for a splicing factor. Throughout stages 1–4 of oogenesis, Caper protein is detected within the nuclei of nurse cells and follicle cells. In stage five and six egg chambers, in addition to expression within nurse cell and follicle cell nuclei, we detect Caper protein within the cytoplasm of the oocyte and within the oocyte nucleus. Furthermore, Caper can be detected in large puncta in the cytoplasm of nurse cells. In stage seven ovarioles and beyond, Caper protein becomes predominantly localized to the oocyte nucleus and is only weakly detected within the oocyte cytoplasm. However, Caper expression remains strong within the nuclei of follicle and nurse cells (Figure 1).

To verify these results, we utilized our hypomorphic *caper* mutant allele, which was generated using a protein trap that results in the fusion of GFP to the Caper protein, and can be used to visualize Caper protein expression [30,37]. Consistent with the expression pattern in *yw* ovarioles stained with an anti-Caper antibody, Caper::GFP is detected in the nuclei of the germarium and within the nuclei of nurse cells and follicle cells throughout oogenesis. While Caper::GFP is also detected within the nucleus of the oocyte, we do not see enrichment of Caper::GFP within the cytoplasm of the oocyte (Figure 1). Since the protein trap results in the inclusion of a GFP encoding exon at the beginning of the Caper protein, it remains possible that the insertion of GFP into the Caper protein disrupts the cytoplasmic localization of Caper. Furthermore, immunoblotting for Caper in ovaries dissected from *caper^−/−^* and *yw* females five days post-eclosion shows that there is not a significant difference in the expression level of Caper in the ovaries between the two genotypes (*caper^−/−^* Caper/alpha-Tubulin ratio = 0.744, *yw* Caper/alpha-Tubulin ratio = 0.828; *t*-Test, t-value = 0.72, P = 0.5114; Appendix A). This suggests that the mechanism of dysfunction in the *caper^−/−^* lines is not the result of differential expression but rather due to the insertion of the GFP coding sequence within the *caper* open reading frame, which likely interferes with protein function. Indeed, while Caper was previously implicated as a splicing factor and would, therefore, be expected to localize to the nucleus, we have shown that Caper also colocalizes with FMRP in the cytoplasm of neurons [31]. Furthermore, two *caper* orthologs, *RBM39/Caperα* and *RBM23/Caperβ* are present in the human genome and show both nuclear and cytoplasmic expression [45,46]. SR proteins, which play roles in the regulation of spliceosome assembly and splice site selection, are known to shuttle between the nucleus and cytoplasm. This shuttling is likely regulated by the phosphorylation of SR proteins. Given that Caper is an SR-like protein, it is possible that its subcellular localization may also be regulated by phosphorylation states. Furthermore, it has been shown using the parasitic nematode *Ascaris lumbricoides*, that SR proteins are hyperphosphorylated and remain within the cytoplasm until the maternal to zygotic transition of embryogenesis. However, upon the initiation of zygotic transcription, SR proteins become partially dephosphorylated and translocate into the nucleus [47].

### 3.2. Caper Dysfunction Results in Lowered Reproductive Output for Females

A previous large scale RNA interference screen for germline stem cell maintenance identified *caper* as important for germline development and germ cell survival. *caper* knockdown was shown to severely reduce the germline in *Drosophila* females [48]. We, therefore, compared ovariole morphology and the reproductive output of *caper*^−/−^ females and controls. We did not detect aberrant ovariole patterning at any stages in *caper*^−/−^ females as compared to controls (Figure 1). We have previously shown that *caper* dysfunction results in several aging phenotypes [31]. Since a decline in reproductive output is also often associated with aging, we investigated whether reproductive output over the lifespan was affected by caper dysfunction. We set up crosses between all combinations of *caper*^−/−^ and *yw* control males and females and then measured the number of eggs laid daily over the lifespan of females. In each of the four genotypic pairings, 25 vials of 5 females each were scored, and females were permitted to continuously mate with males throughout their lifespans. This analysis revealed a significant female x male x day interaction (GLMM: χ^2^ = 35.4, *p* = 2.70 × 10^−9^; Figure 2). Crosses between *caper*^−/−^ females and *caper*^−/−^ males had the lowest reproductive output, and the rate of decline with age was more rapid compared to the other three crosses (Figure 2). The cross between *caper*^−/−^ females and control males also showed an overall reduction in reproductive output relative to crosses involving control females, although the rate of decline with age was similar (Figure 2). Overall, these results indicate that *caper* dysfunction results in reduced female fecundity regardless of the genotype of the male mating partner. Moreover, there appears to be a synergistic interaction between mutant males and females where reproductive output is further reduced in this cross and declines more rapidly with age.

To confirm these results, *caper* was knocked down using RNA interference (RNAi) driven by the germline specific *nanosGal4* driver. *nanosGal4*^+/−^; *UAScaper^RNAi^*^+/−^ females and *nanosGal4*^+/−^ control females were mated to *yw* males, and their reproductive output was recorded over the course of their lifespan. As above, all 125 females utilized for each genotype were provided males to continuously mate with for the duration of the experiment. Similar to *caper* hypomorphic mutant females, *caper* knockdown in the female germline resulted in lower reproductive output compared to controls, and this phenotype was exacerbated by age (GLMM: genotype × day interaction, χ^2^ = 240.0, *p* = 2.20 × 10^−16^; Figure 2).

### 3.3. Caper Dysfunction Results in Partial Embryonic and Larval Lethality

To determine whether *caper* dysfunction might also affect fertilization or embryonic viability, *caper*^−/−^ embryos, *caper*^+/−^ embryos (derived from *caper*^−/−^ and *yw* females mated to *yw* and *caper*^−/−^ males, respectively) were collected, aged 12–24 h and stained with DAPI to determine if eggs were successfully fertilized and the age at which development was arrested. While we did not detect unfertilized eggs in any of the crosses, significantly more embryos from *caper*^−/−^ males and females failed to complete embryonic development when compared to *yw* control and *caper*^+/−^ embryos (GLM, χ^2^ = 50.7, *p* = 5.62 × 10^−11^: Figure 3). In particular, we find that *caper*^−/−^ embryos die significantly more than *yw* embryos (*caper*^−/−^ n = 696, 7.47% dead; *yw* n = 759, 1.71% dead; Tukey’s test: *z*-ratio = −4.9, *p* = 6.58 × 10^−6^) and *caper*
^+/−^ embryos derived from *caper* mutant females outcrossed to *yw* males (n = 793, 3.53% dead; Tukey’s test: *z*-ratio = −3.3, *p* = 0.0055) or *yw* females outcrossed to *caper* mutant males (Tukey’s test: *z*-ratio = −5.4, *p* = 3.45 × 10^−7^). Furthermore, significantly more *caper*^+/−^ embryos derived from *caper* mutant females outcrossed to *yw* males died than embryos derived from *yw* females outcrossed to *caper* mutant males (n = 834, 1.20% dead; Tukey’s test: *z*-ratio = −3.0, *p* = 0.016). Additionally, we found that the majority of embryos died before stage 11, with most *caper*^−/−^ embryos dying between the embryonic development stages four and seven based on the embryonic stages described by Campos-Ortega and Hartenstein [49]. This is of particular interest since these stages correspond to the maternal to zygotic transition (MTZ) when splicing factors begin to remove introns from zygotically expressed genes and are known to change their subcellular localization from the cytoplasm to the nucleus [47,50,51]. We conclude that *caper* is dispensable for fertilization but is partially required for embryonic viability.

Significantly more *caper*^+/−^ embryos derived from *caper* mutant females outcrossed to *yw* males died than embryos derived from *yw* females outcrossed to *caper* mutant males, which suggests a maternal effect for *caper.* Nonetheless, the rate of lethality was not different from *yw* embryos in either of these reciprocal crosses. We, therefore, examined embryonic lethality in embryos derived from *nanosGal4*^+/−^; *UAScaper^RNAi^*^+/−^ females and *nanosGal4*^+/-^ control females mated to *yw* males to better determine if maternal *caper* function is required for embryonic viability. We find that embryos derived from *nanosGal4*^+/-^ control females die 1.6% of the time, whereas embryos derived from females with *caper* knocked down specifically within the germline die 5% of the time (n = 540 for controls and n = 517 for embryos derived from *nanosGal4*^+/−^; *UAScaper^RNAi^*^+/−^ females (GLM: χ^2^ = 9.7, *p* = 0.0019; Figure 3). These results confirm that *caper* maternal function is partly required for embryonic viability.

In some instances, if dysfunction in a gene causes increased lethality at one developmental stage, this effect can also be observed in other stages, as in the cases of the genes *RACK1* and *amontillado* [52,53]. To determine if the partial lethality observed in *caper*^−/−^ and *caper* knockdown embryos was present in any subsequent developmental stages, *caper*^−/−^ and *yw* larvae and pupae were examined and scored for viability in a sex-specific manner. No difference was observed in the survival of either sex at the pupal stage (GLM: genotype × sex, χ^2^ = 2.7, *p* = 0.1007; genotype, χ^2^ = 0.1, *p* = 0.7262; Table 1); however, fewer *caper*^−/−^ female larvae survived to reach pupariation than their control counterparts (Fisher’s exact test: *p* = 0.0006). There was no difference in survival of males through the larval stages (Fisher’s exact test: *p* = 0.2782; Table 1). These results suggest that *caper* is necessary for female survival at the larval stages. However, knockdown of *caper* driven by *ActinGal4* does not result in a significant difference in viability at the larval (GLM: genotype × sex, χ^2^ = 2.9, *p* = 0.0905; genotype, χ^2^ = 1.6, *p* = 0.2035) or pupal stages (GLM: genotype × sex, χ^2^ = 0.2, *p* = 0.6390; genotype, χ^2^ = 1.8, *p* = 0.1825).

### 3.4. Caper Dysfunction Results in a Developmental Delay in Oogenesis

To determine if the lower reproductive output of females with *caper* dysfunction is a result of decreased size in ovaries, we measured the area of ovaries dissected from females at 3, 5 and 14 days post-eclosion. Comparison for the size of ovaries between *caper*^−/−^ and *yw* revealed a genotype by day interaction (LMM, χ^2^ = 14.3, *p* = 0.0008). A genotype by day interaction was also observed when comparing ovary sizes between *nosGal4*, *UASCaperRNAi^GLC01382^* and *nosGal4 ctl* females (LMM, χ^2^ = 49.9, *p* = 1.48 × 10^−11^). Interestingly, at day 3 post-eclosion the ovaries are on average smaller in *caper*^−/−^ females than *yw* females (*caper*^−/−^ n = 50, average = 0.406 mm^2^, *yw* n = 46, average = 0.508 mm^2^, Tukey’s test: *t*-ratio = −2.1, *p* = 0.0352; Figure 4). However, at day 5 post-eclosion there is not a significant difference between the size of the ovaries (*caper*^−/−^ n = 36, average = 0.759 mm^2^, *yw* n = 40, average = 0.782 mm^2^, Tukey’s test: *t*-ratio = −0.4, *p* = 0.6697; Figure 4). Finally, at day 14 post-eclosion the ovaries from *caper*^−/−^ are on average larger than ovaries from *yw* (*caper*^−/−^ n = 44, average = 0.593 mm^2^, *yw* n = 68, average = 0.451 mm^2^, Tukey’s test: *t*-ratio = 3.1, *p* = 0.0020; Figure 4). The progression from being smaller at day 3 to larger at day 14 could be an indication of developmental delay in the *caper*^−/−^ ovaries. However, *caper-*deficient females consistently have lower reproductive output throughout adult stages, with the phenotype being exacerbated with age (Figure 2). Therefore, ovary size may not be the driving factor in the decreased reproductive output observed in aging flies.

When comparing ovary size from females with *nosGal4* driven *UASCaperRNAi^GLC01382^* to those from the *nosGal4 ctl* females, at day 3 the ovaries are on average smaller in the *nosGal4*, *UASCaperRNAi^GLC01382^* females compared to the *nosGal4 ctl* females (*nosGal4*, *UASCaperRNAi^GLC01382^* n = 54, average = 0.666 mm^2^, *nosGal4 ctl* n = 30, average = 0.835 mm^2^, Tukey’s test: *t*-ratio = −4.2, *p* = 4.92 × 10^−5^; Figure 4). The ovaries remain on average smaller in *nosGal4:UASCaperRNAi^GLC01382^* at day 5 (*nosGal4:UASCaperRNAi^GLC01382^* n = 56, average = 0.552 mm^2^, *nosGal4 ctl* n = 50, average = 0.848 mm^2^, Tukey’s test: *t*-ratio = −8.6, *p* = 1.19 × 10^−14^; Figure 4). However, on day 14 there is not a significant difference between the size of the ovaries (*nosGal4:UASCaperRNAi^GLC01382^* n = 48, average = 0.536 mm^2^, *nosGal4 ctl* n = 58, average = 0.491 mm^2^, Tukey’s test: *t*-ratio = 1.3, *p* = 0.1964; Figure 4). The smaller size of ovaries seen in the *nosGal4*, *UASCaperRNAi^GLC01382^* females at days 3 and 5 further supports a developmental delay of oogenesis. However, as previously mentioned, reproductive output in knockdown flies is also exacerbated with age (Figure 2). This again suggests that ovary size may not be the driving factor in reduced reproductive output. We have previously shown that *caper* knockdown generally produces stronger phenotypes than the *caper* hypomorphic allele [31], thus it is not surprising that we see smaller ovaries at both day 3 and day 5 in knockdown animals.

### 3.5. Reproductive Output of Caper^−/−^, but Not Caper Knockdown, Females May Be Impacted by Reduced Matings

Given that mating itself stimulates production of germline stem cells in females [54,55] and decreased mating receptivity could also result in the production of fewer embryos, we next sought to determine if reduced mating factored into the decreased fecundity of *caper*^−/−^ females. To this end, ten chambers containing 15 virgin female and 15 male *caper*^−/−^ or *yw* animals were monitored for two hours for mating. Actively mating animals were removed immediately and scored. Indeed, we found that *caper*^−/−^ females mated significantly less than *yw* females (GLMM: χ^2^ = 10.5, *p* = 0.0012; Table 2). Additionally, significantly more *caper*^−/−^ males were selected as mating partners than *yw* males (GLMM: χ^2^ = 70.9, *p* = 3.83 × 10^−17^). While this may be expected given the established mating defects of both *yellow* and *white* mutant males [56,57,58], our results show that *caper*^−/−^ males do not experience decreased courtship behavior compared to *yw* controls. Thus, lowered reproductive output in crosses including *caper*^−/−^ males may not be attributable to reduced mating initiation by males.

Mating receptivity was next examined in *nanosGal4*^+/−^; *UAScaper^RNAi^*^+/−^ and *nanosGal4*^+/-^ control females to attempt to separate mating behavior from fecundity, since knockdown of *caper* in the female germline should not induce courtship behavioral deficits. Though these females were only mated to *yw* control males in the reproductive output assay described above, *caper*^−/−^ males were provided in the arenas as well to better assess receptivity in general, as we had established that they were preferred by females over *yw* controls. Once again, we observed a similar preference for the *caper*^−/−^ males, as they were selected significantly over the *yw* males (GLMM: χ^2^ = 48.3, *p* = 3.67 × 10^−12^); Table 2. However, there was no significant difference between the number of matings for *caper* knockdown females or controls (GLMM: χ^2^ = 1.0, *p* = 0.3266; Table 2). Thus, while *caper*^−/−^ female reproductive output may be impacted by a reduction in number of matings over their lifespan, reduced fecundity in *caper* knockdown females is likely not the result of fewer matings occurring.

## 4. Conclusions

RNA regulation is integral to the development of the germline and nervous system. Here, we show that the highly conserved RBP Caper is expressed within the female germline throughout all stages of oogenesis. Furthermore, while Caper function is dispensable for oocyte patterning, Caper dysfunction results in decreased reproductive output in *Drosophila* females. Moreover, this phenotype is exacerbated with age. A role for *caper* in germline development may be conserved. For example, in *C. elegans* when the ortholog of *caper*, *rbm-39* is knocked down in a *daf-2* mutant background, there is a significant shortening of the animal’s lifespan compared to *daf-2* single mutants [59]. Additionally, *rbm-39* knockdown in a *lin-35* mutant background results in sterility, pointing to a role for *rbm-39* within the *C. elegans* germline [60]. Given that human and mouse *RBM39* orthologs are also expressed within the female germline, it is likely that *caper* plays a conserved role within the germline.

## Figures and Tables

**Figure 1 jdb-11-00002-f001:**
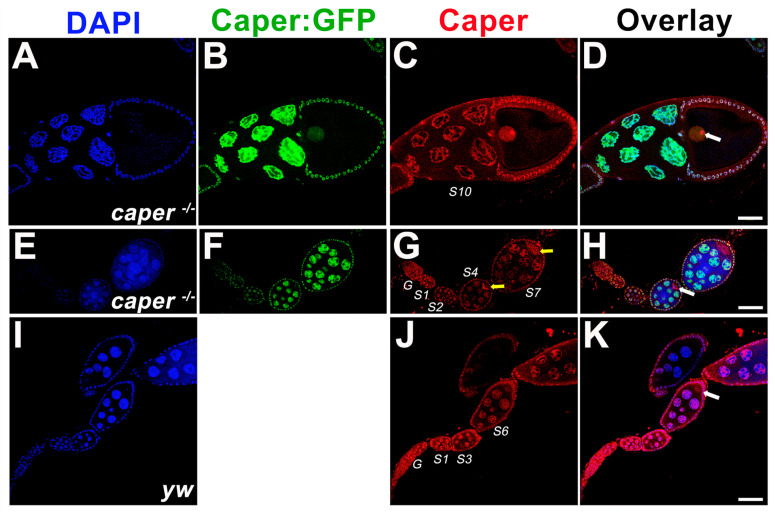
Caper is expressed throughout the female germline. Single z-plane confocal slices indicate that *caper*^−/−^ ovarioles (**A**–**H**) show normal patterning as compared to *yw* ovarioles (**I**–**K**). Nuclei are marked with DAPI in blue (**A**,**E**,**I**). The Caper:GFP fusion protein (**B**,**F**) is shown in green and marks nurse cell nuclei, the oocyte nucleus and follicle cell nuclei. An anti-Caper antibody shown in red (**C**,**G**,**J**) marks nurse cell nuclei, the oocyte nucleus, follicle cell nuclei and is also found in the oocyte cytoplasm. Overlays of all channels are shown in (**D**,**H**,**K**). White arrows mark the oocyte nucleus. Yellow arrows show cytoplasmic expression of Caper marked with an anti-Caper antibody. A 50 micron scale bar is shown in (**K**).

**Figure 2 jdb-11-00002-f002:**
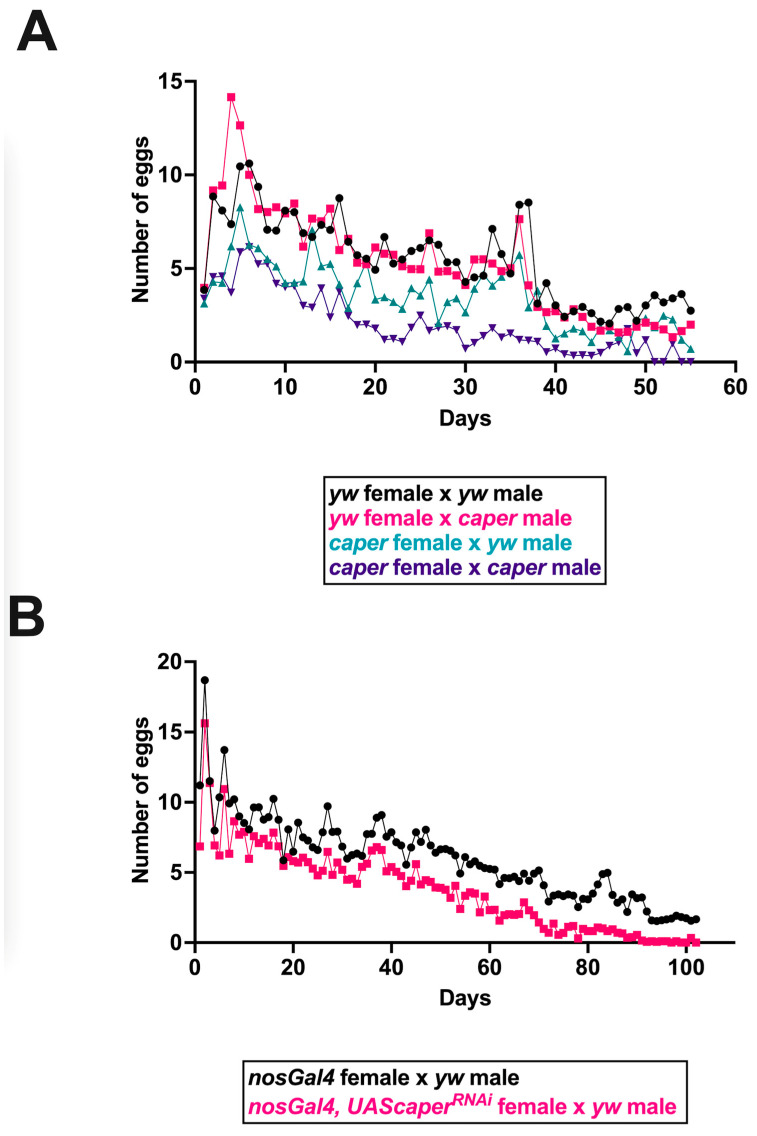
Reproductive output is decreased with *caper* dysfunction. (**A**) Reproductive output is affected in *caper*^−/−^ females regardless of the genotype of the males they are mated with, and these deficits are increased with age. However, *caper*^−/−^ females mated with *caper*^−/−^ males show the greatest reduction in reproductive output. In each of the four genotypic pairings, 25 vials of 5 females each were scored. Statistical analysis of reproductive output revealed a significant female × male × day interaction (GLMM: χ^2^ = 35.4, *p* = 2.70 × 10^−9^). (**B**) Knockdown of *caper* specifically within the germline using *nanosGal4* also results in a decreased reproductive output compared to *nanosGal4* female controls and this phenotype was exacerbated by age (GLMM: genotype × day interaction, χ^2^ = 240.0, *p* = 2.2 × 10^−16^). In both genotypic pairings, 25 vials of 5 females each were scored. The number of eggs laid is plotted on the *y*-axis, with the age of the female in days plotted on the *x*-axis. Genotypes are indicated in the legend for each panel using color coding.

**Figure 3 jdb-11-00002-f003:**
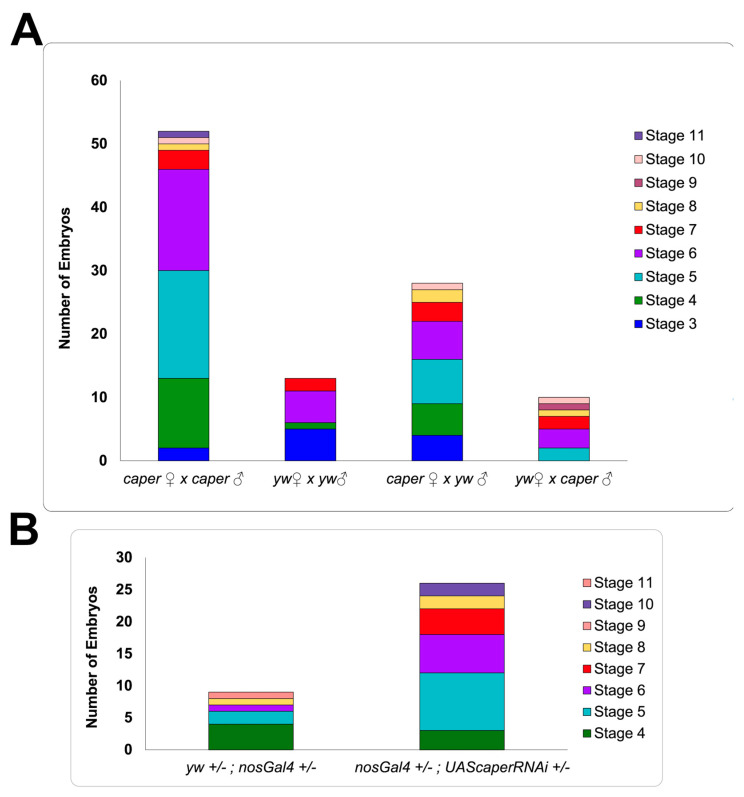
Stages during which embryos arrest development due to *caper* dysfunction. (**A**) The number of embryos that did not complete development and the stage at which they arrested is shown. Number of embryos is plotted on the *y*-axis; parental genotypes are indicated on the *x*-axis. *caper*^−/−^ embryos die significantly more than *yw* embryos (*caper*^−/−^ n = 696, 7.47% dead; *yw* n = 759, 1.71% dead; Tukey’s test: *z*-ratio = −4.9, *p* = 6.58 × 10^−6^) and *caper*
^+/−^ embryos derived from *caper* mutant females outcrossed to *yw* males (n = 793, 3.53% dead; Tukey’s test: *z*-ratio = −3.3, *p* = 0.0055) or *yw* females outcrossed to *caper* mutant males (Tukey’s test: *z*-ratio = −5.4, *p* = 3.45 × 10^−7^). More *caper*
^+/−^ embryos derived from *caper* mutant females outcrossed to *yw* males are embryonically arrested than embryos derived from *yw* females outcrossed to *caper* mutant males (n = 834, 1.20% dead; Tukey’s test: *z*-ratio = −3.0, *p* = 0.016). (**B**) Knockdown of *caper* specifically within the female germline results in a significant increase in the arrest of embryonic development compared to controls (GLM: χ^2^ = 9.7, *p* = 0.00186).

**Figure 4 jdb-11-00002-f004:**
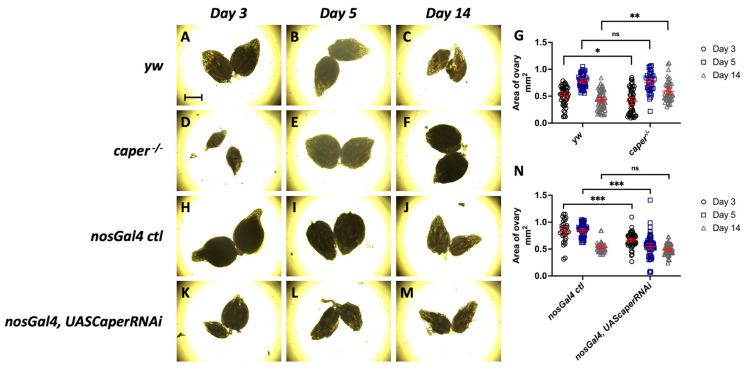
*caper* dysfunction results in smaller ovaries. (**A**–**F**) Images of ovaries dissected from *yw* and *caper*^−/−^ females at days 3, 5 and 14 post-eclosion using a Brightfield microscope. (**G**) Quantification of ovary area shows that ovaries from *caper*^−/−^ females are significantly smaller at day 3 (*caper*^−/−^ n = 50, average = 0.406 mm^2^, *yw* n = 46, average = 0.508 mm^2^, Tukey’s test: *t*-ratio = −2.128, *p* = 0.0352) and significantly larger at day 14 (*caper*^−/−^ n = 44, average = 0.593 mm^2^, *yw* n = 68, average = 0.451 mm^2^, Tukey’s test: *t*-ratio = 3.147, *p* = 0.002) than ovaries from *yw* females. There is no significant difference in ovary size at day 5 (*caper*^−/−^ n = 36, average = 0.759 mm^2^, *yw* n = 40, average = 0.782 mm^2^, Tukey’s test: *t*-ratio = −0.427, *p* = 0.6697). (**H**–**M**) Images of ovaries dissected from *nosGal4 ctl* and *nosGal4*, *UASCaperRNAi^GLC01382^* females at days 3, 5 and 14 post-eclosion using a Brightfield microscope. (**N**) Quantification of ovary area shows that ovaries from *nosGal4*, *UASCaperRNAi^GLC01382^* females are significantly smaller at day 3 (*nosGal4, UASCaperRNAi^GLC01382^* n = 54, average = 0.666 mm^2^, *nosGal4 ctl* n = 30, average = 0.835 mm^2^, Tukey’s test: *t*-ratio = −4.188, *p* = 4.92e × 10^−5^) and day 5 (*nosGal4:UASCaperRNAi^GLC01382^* n = 56, average = 0.552 mm^2^, *nosGal4 ctl* n = 50, average = 0.848 mm^2^, Tukey’s test: *t*-ratio = −8.619, *p* = 1.19 × 10^−14^) than ovaries from *nosGal4 ctl*. At day 14 there is not a significant difference in ovary size (*nosGal4:UASCaperRNAi^GLC01382^* n = 48, average = 0.536 mm^2^, *nosGal4 ctl* n = 58, average = 0.491 mm^2^, Tukey’s test: *t*-ratio = 1.298, *p* = 0.1964). Scale bar on panel A represents 500 μm.Lines within each graph represent the mean and 95% confidence interval and significance is indicated by * *p* ≤ 0.05, **, *p* ≤ 0.01, *** *p* ≤ 0.001 or ns (not significant).

**Table 1 jdb-11-00002-t001:** *caper* is required for viability during larval stages but is dispensable during pupariation.

	Males				Females			
	*yw*	*caper* ^−/−^	*RNAi Control*	*caper RNAi*	*yw*	*caper* ^−/−^	*RNAi Control*	*caper RNAi*
Larval Death	3(63)	5(46)	0(46)	3(43)	0(59)	9(50)	2(53)	2(49)
Pupal Death	6(60)	8(41)	2(46)	1(40)	8(59)	3(41)	4(51)	1(47)

Cells are formatted A(B), such that A represents the number of animals that died during the developmental stage shown, while B shows the total number of animals examined. *RNAi control* describes the *ActinGal4* outcrossed to *yw*, while *caper RNAi* describes the *ActinGal4*; *UAScaperRNAi^HMC^* animals.

**Table 2 jdb-11-00002-t002:** *caper* dysfunction results in decreased mating rates for females but not males.

		Female Genotype				
		*yw*	*caper* ^−/−^	*yw*; *nosGal4*	*nosGal4*; *UAScaperRNAi*	Totals
**Male Genotype**	*yw*	6	0	2	4	270
	*caper* ^−/−^	60	39	38	29	270
	Totals	150	150	120	120	

*yw* males mated significantly less than *caper*^−/−^ males in both *caper*^−/−^ (*p* = 1.45 × 10^−33^) and *caperRNAi* (*p* = 1.04 × 10^−19^) trials. While *caper*^−/−^ females mated significantly less than *yw* control females (*p* = 0.001), there was no significant difference in mating between *caperRNAi* females and their *nosGal4* outcross control (*p* = 0.33). Numbers shown indicate the number of successful matings for a given cross.

## Data Availability

Not applicable.

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
