# Peer review of "Roles for the RNA-Binding Protein Caper in Reproductive Output in Drosophila melanogaster"

_jdb, 2022, doi:10.3390/jdb11010002_

Round 1
Reviewer 1 Report
This manuscript reported RNA-binding protein caper has significant function in oogenesis and fertility. This is a quite interesting finding, but the authors did only a few experiments and thus raised more questions than answers.
1. Basically, Fig.1 and Fig.2 are the main results of this manuscript. Fig.3 showed the stages of the developmental arrest, which is not in the scope of oogenesis and fertility. It seems that the authors wrote in the introduction that it is known that the caper is functional during the embryogenesis.
2. No statistics was performed in the Fig.2. The authors should do good statistics and even more biological replicates for firm conclusions.
3. caper -/- produced less eggs, seemingly (Fig. 2A). But how? At which stage of oogenesis? The authors gave only a very descriptive picture.
4. Since caper is an RNA binding protein, the authors should at least perform a RIP-seq to see what RNAs are bound by caper and thus are influenced in translation (for example).
5. Figure legends are too simple and not informative (Fig.2, 3)
Author Response
This manuscript reported RNA-binding protein caper has significant function in oogenesis and fertility. This is a quite interesting finding, but the authors did only a few experiments and thus raised more questions than answers.
- Basically, Fig.1 and Fig.2 are the main results of this manuscript. Fig.3 showed the stages of the developmental arrest, which is not in the scope of oogenesis and fertility. It seems that the authors wrote in the introduction that it is known that the caper is functional during the embryogenesis.
We specifically performed these experiments to determine whether eggs were being fertilized. Given fertilization was not the issue, we looked at the stage of developmental arrest.
- No statistics was performed in the Fig.2. The authors should do good statistics and even more biological replicates for firm conclusions.
We have performed robust statistics for all experiments and the description is found in materials and methods. We also added the statistical significance to figure legends. Furthermore, we have many biological replicates for these experiments. 125 females were examined for each of the different genotypic matings, providing a robust n value for these experiments.
“Statistical Analyses
Fecundity data was analyzed using a generalized linear mixed model (GLMM) with a negative binomial error distribution (NB1 parameterization) using the R package glmmTMB (Brooks et al., 2017). The full factorial model included female line, male line, and day as factors. Replicate was treated as a random effect to account for repeated measurements from the same set of flies over time. Model fit was assessed using diagnostic plots generated by the R package DHARMa (Hartig, 2017). An anova table was generated from model results using the R package car (Fox & Weisberg, 2019). Mortality at different developmental stages was analyzed using GLMs with a binomial error distribution and logit link function. However, the experiment comparing larval survival of caper-/- and controls was analyzed using Fisher’s exacts tests because the fact that no female control larvae died hindered estimation of the glm model due to quasi-complete separation. The glm model for embryo mortality included cross as a factor, while larval and pupal mortality analyses included both cross and sex as factors as well as the interaction between these variables. Anova tables were generated from model results using the R package car (Fox & Weisberg, 2019). Post-hoc comparisons were analyzed with the R package emmeans using Tukey’s adjustment (Lenth 2021). Male and female mating frequency was analyzed by GLMMs with a binomial error distribution and logit link function using the R package lme4 (Bates et al., 2015). Replicate was treated as a random variable. Ovary size data was analyzed with a full factorial linear mixed effects model using the R package lme4. The model included factors for genotype and day and their interaction. Fly id was treated as a random effect since both ovaries were measured for each fly. Anova tables were generated from model results using the R package car (Fox & Weisberg, 2019). Post-hoc comparisons were analyzed with the R package emmeans using Tukey’s adjustment (Lenth, 2021). Ovary size was analyzed by GLMMs using the R package lme4 (Bates et al., 2015). Which fly the ovary came from was treated as a random variable to account for two ovaries coming from each fly. “
All statistics are presented within the results section of the manuscript. Due to the nature of the statistical tests performed, and the number of post-hoc analyses it is not possible to simply display the results of statistical analyses within the figures. Therefore, the results of the analyses are within the text of the results section of the manuscript.
- caper -/- produced less eggs, seemingly (Fig. 2A). But how? At which stage of oogenesis? The authors gave only a very descriptive picture.
Individually dissecting ovaries and staging all of the egg chambers is not trivial. Generally, in the process of dissection, transfer and mounting, there is significant concern of losing certain stages and if tissue overlaps it can be difficult to stage egg chambers accurately. Furthermore, with hundreds of egg chambers per ovary, to get a sufficient n value is extremely difficult. That said, our expression analyses show that all stages of ovarian development are represented in our mutant lines and the overall patterning of ovaries is normal, as stated in the manuscript.
To address this, we have, however, dissected out ovaries and measured their overall size in yw controls, caper mutant animals, as well as in knockdown animals and their respective controls.
Our results indicate that ovaries derived from 3-day old caper mutant female virgins are significantly smaller in area than their respective controls. Furthermore, knockdown animals also show significantly smaller area of the ovary compared to controls on both day 3 and day 5. This data has been added to the manuscript and to the figures.
- Since caper is an RNA binding protein, the authors should at least perform a RIP-seq to see what RNAs are bound by caper and thus are influenced in translation (for example).
This is well beyond the scope of this manuscript. We have a separate manuscript under review where we perform RIP Seq for Caper as well as co-IP followed by LC-MS to identify Caper interacting proteins.
- Figure legends are too simple and not informative (Fig.2, 3)
We have rewritten the figure legends to provide more specific details regarding n values and statistical analyses.
Reviewer 2 Report
In the manuscript entitled “Roles for the RNA-Binding Protein Caper in Reproductive Output,” the authors have shown the novel roles of Caper in fertility and mating behavior. The results demonstrate that RNA Binding Protein Caper is expressed in ovarian follicles throughout oogenesis. Additionally, reduced caper function leads to a decreased reproductive output in Drosophila females. Moreover, this phenotype is exacerbated with age. The authors have also shown that caper dysfunction results in partial embryonic and larval lethality. Given that Caper is highly conserved across metazoan, the study may have future implications in the field.
The results are interesting, and the findings advance the reproductive and developmental biology areas. However, some technical issues must be addressed to support the paper's conclusions, which are outlined below.
· In Figure 1, the authors have shown the expression of Caper in caper mutants as well as in yw controls. Though, the authors have hypothesized why the cytoplasmic localization of Caper could not be visualized in Caper: GFP lines (Figure 1F), I am curious why one could not see the exact expression of Caper with Caper antibody (Figure 1B and C). Authors could replace the image with a better picture.
· The authors have shown in their results that caper hypomorphs show Caper expression. Is it equivalent to wild type? The authors could provide quantification data for the same. Moreover, RT-PCR/Western Blot data could have supported their conclusion in a better way.
· It would be good if the authors could present the ovarioles in a stage-specific manner or at least this should be labeled in the figure itself.
· In figure 2, # of eggs should be replaced with the “Number of eggs”.
· The manuscript conclusion is sometimes too speculative. The authors state, “Since both locomotor dysfunction and decreased fecundity are common measures of aging phenotypes in Drosophila, it is likely that caper is a general regulator of aging”. Claiming Caper as a general regulator of aging is an overstatement with the data presented. When suggesting speculative working models and/or hypotheses the authors should write those possibilities as such, and not as conclusions.
Author Response
In the manuscript entitled “Roles for the RNA-Binding Protein Caper in Reproductive Output,” the authors have shown the novel roles of Caper in fertility and mating behavior. The results demonstrate that RNA Binding Protein Caper is expressed in ovarian follicles throughout oogenesis. Additionally, reduced caper function leads to a decreased reproductive output in Drosophila females. Moreover, this phenotype is exacerbated with age. The authors have also shown that caper dysfunction results in partial embryonic and larval lethality. Given that Caper is highly conserved across metazoan, the study may have future implications in the field.
The results are interesting, and the findings advance the reproductive and developmental biology areas. However, some technical issues must be addressed to support the paper's conclusions, which are outlined below.
- In Figure 1, the authors have shown the expression of Caper in caper mutants as well as in yw controls. Though, the authors have hypothesized why the cytoplasmic localization of Caper could not be visualized in Caper: GFP lines (Figure 1F), I am curious why one could not see the exact expression of Caper with Caper antibody (Figure 1B and C). Authors could replace the image with a better picture.
We have replaced this image with a better image. Furthermore, we included an excerpt explaining that the reason why we may not see complete co-localization of the GFP and the Caper antibody is that inclusion of the GFP in the protein may be disrupting cytoplasmic localization.
- The authors have shown in their results that caper hypomorphs show Caper expression. Is it equivalent to wild type? The authors could provide quantification data for the same. Moreover, RT-PCR/Western Blot data could have supported their conclusion in a better way.
To address this we have performed Western Blot analyses on ovarian tissue. We chose to perform Western blotting instead of RT-PCR because the caper locus encodes multiple poison exon isoforms and therefore RNA levels may be misleading with respect to Caper protein levels.
We find that the levels of Caper expression are not changed between the mutant and control lines. This is not necessarily surprising, as this is a GFP trap line, where the insertion of GFP into the open reading frame is likely interfering with Caper function.
- It would be good if the authors could present the ovarioles in a stage-specific manner or at least this should be labeled in the figure itself.
The stages have now been marked in the figure.
- In figure 2, # of eggs should be replaced with the “Number of eggs”.
This has been corrected
- The manuscript conclusion is sometimes too speculative. The authors state, “Since both locomotor dysfunction and decreased fecundity are common measures of aging phenotypes in Drosophila,it is likely that caper is a general regulator of aging”. Claiming Caper as a general regulator of aging is an overstatement with the data presented. When suggesting speculative working models and/or hypotheses the authors should write those possibilities as such, and not as conclusions.
We have tempered this language within the conclusions section and removed any statements regarding a role for Caper in aging.
Round 2
Reviewer 1 Report
I thank the authors to perform additional experiments and analyses to address my questions. Basically, my major concerns have been addressed.
Still, minor changes are needed. For example, the chi-square should not be written as X2, but using greek letter. The "P=" were not consistent over the manuscript (capitalized, italic, ...) . Very small P-values should be written as 3.83x10-17 or 3.83E-17, not 3.83e-17. The type of statistical test should be mentioned whenever a P-value was mentioned.
Besides, I strongly recommend the authors to change the color scheme in Fig.4. The current colors do not represent any stage ranks. Using rainbow colors may indicate the fraction of early/mid/late stages visually. Otherwise, the authors may also choose to mark the stage number next to ever color blocks in the figure.
The panel letters in Fig.4 are very difficult to follow. The panels G and N are also very difficult to follow. Due to the very dense data points, the readers cannot distinguish the shape of the points. Therefore, the Day3/5/14 should be directly written in the diagram, not as figure legends.
Author Response
I thank the authors to perform additional experiments and analyses to address my questions. Basically, my major concerns have been addressed.
Still, minor changes are needed. For example, the chi-square should not be written as X2, but using greek letter. The "P=" were not consistent over the manuscript (capitalized, italic, ...) . Very small P-values should be written as 3.83x10-17 or 3.83E-17, not 3.83e-17. The type of statistical test should be mentioned whenever a P-value was mentioned.
- We note that we are using the Greek letter in advance symbols in Word. We have made all other recommended changes.
Besides, I strongly recommend the authors to change the color scheme in Fig.4. The current colors do not represent any stage ranks. Using rainbow colors may indicate the fraction of early/mid/late stages visually. Otherwise, the authors may also choose to mark the stage number next to ever color blocks in the figure.
- We think the reviewer is referring to Figure 3 and not 4. But we are not understanding what is being requested because the graph is already in rainbow colors and the stages are indicated in the legend. Is there possibly a problem with opening the figure file such that it is not displaying in color?
The panel letters in Fig.4 are very difficult to follow. The panels G and N are also very difficult to follow. Due to the very dense data points, the readers cannot distinguish the shape of the points. Therefore, the Day3/5/14 should be directly written in the diagram, not as figure legends.
- We have made the panel letters much larger. We have also changed the graphs such that each day is a different color to make following the graph easier.